# Microbiological Evaluation of Novel Bis-Quaternary Ammonium Compounds: Clinical Strains, Biofilms, and Resistance Study

**DOI:** 10.3390/ph15050514

**Published:** 2022-04-22

**Authors:** Nikita Frolov, Elena Detusheva, Nadezhda Fursova, Irina Ostashevskaya, Anatoly Vereshchagin

**Affiliations:** 1N. D. Zelinsky Institute of Organic Chemistry, Russian Academy of Sciences, Leninsky Prospect 47, 119991 Moscow, Russia; nfrolov@ioc.ac.ru (N.F.); ostashevskaia@ioc.ac.ru (I.O.); 2State Research Center for Applied Microbiology and Biotechnology, Obolensk, 142279 Serpukhov, Moscow Region, Russia; DetushevaEV@obolensk.org (E.D.); fursova@obolensk.org (N.F.); 3Faculty of Chemistry, Moscow State University, Leninskie Gory 1, 119991 Moscow, Russia

**Keywords:** quaternary ammonium compounds, antiseptics, disinfectants, biocides, pyridinium salts, antibacterial activity, antibiofilm activity, bacterial resistance

## Abstract

This work is devoted to the investigation of biocidal properties of quaternary ammonium compounds (QACs) based on pyridine structures with aromatic spacers, and their widely known analogs, against clinically significant microorganisms. This study is focused on investigating their antimicrobial activity (minimum inhibitory concentrations (MICs) and minimum bactericidal concentrations (MBCs)), antibiofilm properties (minimum biofilm inhibitory concentrations (MBICs) and minimum biofilm eradication concentrations (MBECs)), synergetic effect with different alcohols in antiseptic formulations, and bacterial resistance development. It was shown that all combined analogue preparations had a higher level of antibacterial activity against the tested bacterial strains, with a 16- to 32-fold reduction in MICs and MBCs compared to previously used antiseptic preparations. Moreover, hit-QACs demonstrated a stable effect against Gram-negative *E. coli*, *K. pneumoniae,* and *A. baumannii* within a month of incubation. Overall results indicated a high level of antibacterial activity of pyridine-based QACs.

## 1. Introduction

In recent years, the world has been threatened by the spread of infectious diseases due to ever-increasing pathogenic microorganisms’ resistance to widely used antibacterial drugs [1]. The treatment of infections associated with biofilms is especially difficult. Bacteria in biofilms are known for higher survival rate in the presence of aggressive substances, immune defense factors, and antibacterial drugs compared to free-floating planktonic cells [2,3]. Therefore, standard antibiotic treatment can only destroy planktonic cells without affecting attached forms, which can survive in biofilms and multiply when treatment is completed [4]. It is necessary to use large doses of expensive drugs for a long time in order to achieve bactericidal concentrations of antibiotics during the treatment of such infections. The main drawback is a significant risk of various side effects.

One of the ways to solve the problem is developing new drugs with targeted local antimicrobial activity—modern antiseptic drugs. Nowadays, the most promising components of antiseptics are quaternary ammonium compounds (QACs). Currently, QACs have a wide range of applications: they are used as therapeutic and prophylactic antiseptics, preservatives for medications and cosmetics, disinfectants and surface clearances, deodorants, etc. [5]. These compounds have high activity rate against a wide range of pathogens, such as bacteria, fungi, and some viruses. Along with an antimicrobial effect, QACs can be used as detergents, which allows combining disinfection with cleaning. They are not volatile and are not dangerous when inhaled [6]. QACs belong to the class of cationic biocides [7]. The main target of their action is the bacterial lipid bilayer [5]. However, the nature of QACs’ mechanism is a matter of speculation [8]. Thus, QAC-based disinfectants have low toxicity, good detergent properties, and high bactericidal efficacy against Gram-positive and Gram-negative bacteria.

A significant number of QACs have been obtained throughout the last century [9,10]. At the same time, the practical use of QAC-based disinfectants requires constant epidemiological and microbiological monitoring due to microbial adaptation [11,12,13,14]. In this regard, the requirement for development of new highly active chemical compounds and microbiological investigations increases. Previously, we published several articles about the discovery of new highly active bis-pyridinium QACs with aromatic spacers (Figure 1) [15,16]. Hit compounds showed great inhibitory activity against a broad spectrum of planktonic bacteria and fungi. In this study, we expanded microbiological research of our most active QACs. Thus, QACs were tested on clinical multi-resistant strains (including research of resistance development), and bacterial biofilms. Moreover, new antiseptic compositions have been developed using combinations of novel QACs and different alcohols. Minimal inhibitory concentrations (MICs) and minimal bactericidal concentrations (MBCs) as well as minimum biofilm inhibitory concentrations (MBICs) and minimum biofilm eradication concentrations (MBECs) were estimated for all tested samples.

## 2. Results and Discussion

### 2.1. Intensity of Biofilm Formation

The ability to form biofilms was estimated for each bacterial strain. The intensity of biofilm formation was determined by density analysis. According to the results, bacterial strains were divided into two categories: the strains with a high rate of biofilm formation (*n* = 6) and those with a modest one (*n* = 4). Worth noting is that clinical pathogens were more prone to forming biofilms. Thus, four out of five clinical strains showed a high OD_590_ intensity, which was more than 4 times greater compared to the control strain (Table 1 and Table 2).

### 2.2. Antibacterial Activity of Tested QACs

At first, we re-examined previously obtained hit compounds **1**–**3** with a suspension method and got different results from our studies in collaboration with CO-ADD (Institute for Molecular Bioscience, Brisbane, Australia) [15,16]. Modern popular commercial QACs (benzalkonium chloride (**BAC**), cetylpyridinium chloride (**CPC**), and octenidine dichloride (**OCT**)) were used as reference compounds (Figure 1). Detailed information on the experimental procedures and bacterial strains used is represented in the experimental section.

The microbiological study showed that all tested QACs **1**–**3** had a high level of antibacterial activity against Gram-positive and Gram-negative planktonic bacteria with slightly different MIC from previous research (Table 3). However, this did not oppose the general picture.

The vast majority of the tested biofilms were more resilient to the studied compounds than the same bacterial strains in planktonic form (Table 3). Apparently, this was caused by the influence of different factors and characteristics of bacterial biofilms—the presence of an extracellular matrix [17], properties of substrate in which biofilms are formed [2], and production of resistance genes in biofilms [4,18]. The highest level of resistance was noted for *A. baumannii* and *P. aeruginosa*. This can be justified by specific features of these bacterial strains and it is consistent with the results of our previous studies [19]. It should be noted that Gram-positive *S. aureus* were more sensitive to the studied QACs than Gram-negative bacteria. Commercial mono-QAC **BAC** had extremely low rate of antibacterial activity against bacterial strains, in both planktonic (Table 2) and biofilm (Table 3) forms. For **OCT,** high antibacterial activity against planktonic cells was noted (Table 2). However, **OCT** showed a much lower level of biocide action against biofilms. Surprisingly, the activity of the studied compounds against clinical strains did not differ much from the reference cultures, and in some cases was even more expressed. Overall, the effectiveness of new bis-QACs **1-3** was higher than the effectiveness of reference antiseptics **BAC** and **CPC** and was relatively comparable to **OCT** (Table 2).

### 2.3. Antiseptic Compositions

Next, we investigated the combined effect of new QACs with various alcohols in antiseptic compositions. Alcohols are widely known as antibacterial agents. Those are used in many antiseptics and disinfectants as active substances [20], in cosmetic products as preservatives [21], etc. Alcohol-based hand sanitizers are recommended by WHO as preventive measures against COVID-19 [22]. Moreover, some studies report the increase in antibacterial activity when usage of biocides and alcohols is combined [23,24,25].

We chose isopropyl alcohol, propyl alcohol, and phenoxyethanol for various biocidal formulations. The first two are the most popular on the market and phenoxyethanol is used in combination with **OCT**, which is the antiseptic of choice for wound treatment according to European guidelines [26]. We prepared the following **formulations**: (1) QAC + phenoxyethanol/water (2% *v/v* solution) by analogy with the drug “Octenisept” (Schülke & Mayr) − **PhE**; (2) QAC + isopropyl alcohol/water (63.5% *v/v* solution) − **IPA**; (3) QAC + isopropyl alcohol/propyl alcohol/water (1:2:2 by volume) − **IPAP**. **OCT** and **BAC** were used as standards in the same formulations. “Empty” (without QACs) compositions were used as control samples (Table 4 and Table 5).

Based on the data obtained, a number of tendencies were noted. In most cases, we observed a biocide effectiveness increase for antiseptic formulations of new QACs (**1–3**) in combination with alcohols. The most active of all compositions was QACs with 63.5% (*v/v*) **IPA** (isopropanol) solution, that was slightly superior in antibacterial properties to the combination with a mixture of isopropanol and propanol-1 (**IPAP**). Particular attention should be paid to the considerable increase in activity against *A. baumannii* and *P. aeruginosa*, belonging to the ESCAPE pathogens group, the most common cause of nosocomial infections, highly resistant to the majority of antimicrobial drugs. The MIC on *A. baumannii* planktonic cells for **IPAP 1** was 32-fold less than for individual QAC **1**, and MBC was 16-fold less (Table 2 and Table 4). **IPA 1** showed superior properties against *P. aeruginosa* planktonic cells and biofilms, with a 16-fold reduction in MBC against both bacterial forms. Phenoxyethanol (**PhE**) at the concentration used was less effective. Thus, the MIC and MBC values of new QACs did not differ from those of individual substances in ~35% of all cases. For comparison, **IPA** had ~15% with no biocidal rate improvement and **IPAP** ~22%. It is worth noting that all alcohols enhanced the effect of QACs on *S. aureus* biofilms in 99% of cases. The summary table of the overall antiseptic compositions’ efficacy is represented below (Table 6).

### 2.4. Bacterial Resistance Study

Furthermore, we investigated formation of bacterial resistance to new QACs **1–3** and the best of the commercial antiseptics **OCT**. A complete description of the research methodology is presented in the experimental section. The results obtained are plotted as MBC vs. incubation time of biocides with bacterial cells. The output of the graph to a plateau corresponds to the end of the experiment and indicates the cessation of resistance development by microorganisms (Figure 2).

The Gram-positive strain of *S. aureus* developed resistance against all tested samples after 20–25 days of incubation. MBCs of compounds **1**, **3**, and **OCT** increased 8-fold, and for the compound **2**-**4**-fold. However, **OCT** activity remained the highest after 40 days with an MBC value of 4 mg/L. The Gram-negative *E. coli* strain developed resistance to commercial QAC, increasing the MBC 64-fold after 25 days of treatment. On the other hand, for compounds **2** and **3**, MBCs increased only 2-fold (up to 16 mg/L and 32 mg/L). Moreover, for compound **1**, resistance did not appear at all after 25 days (Figure 2). *K. Pneumoniae* had developed resistance to all QACs during the first 10 days of incubation. MBC of **OCT** increased from 8 mg/L to 125 mg/L within a month. For new bis-QACs **1** and **3**, MBCs increased 4-fold, reaching 32 mg/L and 63 mg/L, respectively. Compound **2** retained MBC at 16 mg/L after a month of the experiment, which is 2 times higher than the original value (Figure 2). In compounds **1** and **3**, after 40 days of incubation with the *A. baumannii*, MBCs increased only 2-fold to 32 mg/L and 16 mg/L, respectively. Bis-QAC **2** did not induce resistance, maintaining its activity at 16 mg/L throughout the entire study (Figure 2). **OCT** was not tested against *A. baumannii* with an MBC value of 125 mg/L (Table 3) due to poor biocidal properties. *P. aeruginosa* was proved to be the most resilient among all the studied strains. Thus, all bis-QACs lost their activity during the first 20 days of the experiment (Figure 2). Overall, new QACs showed great potential in the stability of bactericidal properties on Gram-negative strains over the time compared to **OCT**, which, surprisingly, had a dramatic loss of efficacy against the bacteria tested. These results indicate that aromatic moieties in bis-QACs’ spacer may cause less substrate recognition by resistance system in Gram-negative bacteria than Gram-positive ones. However, this statement requires further investigations.

## 3. Materials and Methods

### 3.1. Bacterial Strains

Reference strains of microorganisms *E. coli* ATCC 25922, *K. pneumoniae* ATCC 70060, *S. aureus* ATCC 43300, *P. aeruginosa* ATCC 27853, and *A. baumannii* ATCC 15308 were obtained from the State Collection of Pathogenic Microorganisms (Obolensk, Russia). Clinical strains of *E. coli* B-3421/19, *K. pneumoniae* B-2523/18, *S. aureus* B-8648, *P. aeruginosa* B-2099/18, and *A. baumannii* B-2926/18 were isolated in the Molecular Microbiology department of the Federal Budget Institution of Science State Research Center for Applied Biotechnology and Microbiology (FBSI SRC PMB, Obolensk, Russia) from clinical samples in the investigation of infection cases in 2016–2018.

### 3.2. Identification of Microorganisms

Species identification of microorganisms was carried out on a MALDI-TOF Biotyper mass spectrometer (Bruker, Billerica, MA, USA). One part of daily bacterial culture was introduced into a 2 mL tube (Eppendorf, Hamburg, Germany) containing 300 µL of deionized water and was triturated until a homogeneous suspension was obtained. Then, 900 µL of 96% ethanol was added and shaken. The tube was then centrifuged at 12,000× *g* within 2 min. The supernatant was discarded, and the precipitate was dried in air. After drying, the precipitate was thoroughly mixed with 30 µL of 70% formic acid, incubated for 10 min at room temperature, then mixed with 30 µL of acetonitrile (Sigma-Aldrich, St. Louis, MO, USA) and incubated for another 10 min. This mixture was centrifuged at 12,000× *g* for 2 min; then, 1 µL of the supernatant was transferred to a MSP 96 Polished Steel MALDI Target Plate (Bruker Daltonik GmbH, Bremen, Germany), air-dried, and covered with 1 µL of a saturated solution of α-cyano-4-hydroxycinnamic acid (Bruker Daltonik, Bremen, Germany). The MALDI target plate was placed in a MALDI-TOF microflex LRF instrument (Bruker Daltonik, Bremen, Germany). The data acquisition settings were as follows: ion source 1 at 19.50 kV, ion source 2 at 18.22 kV, lens at 7.01 kV, and a mass range of 2000 to 20,000 Da. Spectrum registration was performed automatically using the MALDI Biotyper RTC 3.1 software (Bruker Daltonik, Bremen, Germany). The obtained spectra were compared with the reference spectra of the FFL v1.0 database. Identification scores ≥2.0 and 1.7–1.99 indicated recognition of the species and genus levels, respectively, while scores <1.7 indicated a lack of reliable data.

### 3.3. Antibacterials

All investigated QACs were synthesized in N. D. Zelinsky Institute of Organic Chemistry Russian Academy of Sciences. Full experimental and characterization data can be found elsewhere [15,16]. Phenoxyethanol and benzalkonium chloride were purchased from Acros Organics and used without further purification. Cetylpyridinium chloride was synthesized from pyridine using the simple method of alkylation in acetonitrile. Octenidine dihydrochloride was synthesized using previously described method [27]. Propanol-1 and propanol-2 were purchased from Rushim and purified using distillation.

### 3.4. Cultivation of Microorganisms

Microorganisms were cultivated using agar and GRM broth. Cultivation of microorganisms was carried out for 20–24 h at a temperature of 37 °C.

### 3.5. Determination of the Biofilm Formation Ability

The efficiency of bacterial biofilm formation was determined using a method based on the ability of the crystal violet dye to bind with cells and biofilm matrix [28]. To obtain biofilms, 96-well flat-bottomed culture plates were used. In each well, 200 µL of a daily bacterial culture was inoculated at a concentration of 10^6^ CFU/mL and cultivated for 24 h at a temperature of 37 °C. Then, the medium with planktonic cells was carefully taken from the plate. Wells with biofilms were washed for 2–3 min using sterile PBS buffer (NaCl—8 g, KCl—0.2 g, Na_2_HPO_4_—1.44 g, KH_2_PO_4_—0.24 g per 1 L, pH = 7.4) in the same volume to remove the residual planktonic cells. After washing, PBS was completely removed by pipetting. Then, 200 µL of a filtered 0.1% solution of gentian violet was added to each well, and the biofilms were incubated with the dye for 10–15 min at room temperature. The dyes were completely removed from the well by pipetting. The unbound dye was thoroughly washed off with PBS. The plates were inverted onto filter paper and dried. After complete drying of the surface, a 200 μL mixture of ethanol–isopropanol (1:1) was added to the wells. The dye was washed off from the surface of the wells, taken, and placed in clean flat-bottomed plates. The optical density of the resulting solution was measured at a wavelength of 590 nm using an xMark™ Microplate Absorbance Spectrophotometer (Bio-Rad Laboratories Inc., Hercules, CA, USA). The measurement results were interpreted by comparing the OD_590_ values of the samples with control ones (pure solvent without dye added).

The absence of biofilm was recorded at OD_590_ sample ≤ OD_590_ control; weak degree of biofilm production at OD_590_ control < OD_590_ sample ≤ 2*OD_590_ control; average degree of biofilm production at 2*OD_590_ control < OD_590_ sample ≤ 4*OD_590_ control; high degree of biofilm production at OD_590_ control < 4*sample OD_590_; in accordance with the recommendations of Rodrigues et al. [29]. All experiments were carried out in triplicate.

### 3.6. Antibacterial Assay

MIC and MBC were determined by the suspension method. For this, two-fold dilutions of the tested solutions (500–0.25 mg/L) were prepared in nutrient broth. The nutrient broth with tested QACs (0.1 mL) was added to 12 wells in the horizontal rows of the culture plate. In separate rows, nutrient broth without QACs was added for control measures. In the case of antibiofilm analysis, nutrient broth with the appropriate concentration of the tested compounds was added in 0.1 mL to 12 wells in horizontal rows of a culture plate with washed biofilms, which were prepared according to the abovementioned method.

From single colonies grown on GRM medium at 37 °C for 18 h, a suspension was prepared with an optical density of 0.5 according to the McFarland standard in sterile saline, which corresponds to approximately 1–2 × 10^8^ CFU/mL. The suspension was then diluted 100-fold by adding 0.2 mL of the suspension to a flask containing 19.8 mL of Mueller–Hinton broth (MHB). The concentration of microorganisms, in this case, was 10^6^ CFU/mL. An amount of 0.1 mL of the initial suspension was added to the wells with the test drug and control wells with broth. The final concentration of the microorganisms in each well was 5 × 10^5^ CFU/mL. The plates were covered with lids and placed in a thermostat (37 °C) for 20 h. The presence of bacterial growth was taken into account visually (according to the presence of turbidity in the well). MIC was taken to be the minimum concentration of the preparation at which bacterial growth was absent after 20 h of incubation. MBC was determined by the results of inoculation on dense nutrient media. To do this, from all the wells in which there was no visible growth (according to the presence of turbidity), 10 μL were sown on Mueller–Hinton nutrient agar. The results were taken into account by the presence of culture growth at the site of application after 24 h of incubation at a temperature of 37 °C. If there was no growth in the well, but the growth of the studied culture was observed when seeding from this well on a solid nutrient medium, then this concentration was taken as MIC. The lowest concentration was taken as MBC, at which cell growth was completely suppressed when seeded on a dense nutrient medium.

### 3.7. Bacterial Resistance Assay

The selection of resistant microorganisms is carried out by regular culture of the bacterial strains in a nutrient broth containing serial dilutions of the antibacterial drug. To do this, in 2 mL of nutrient broth, a series of two-fold dilutions of the drug were prepared, starting with a concentration that was half the MIC. During the initial inoculation, 20 μL of 107 CFU/mL of daily culture was added to each tube and incubated at 37 °C. After two to three days of incubation, 50 µL from the last turbid tube with a higher concentration was divided into new ones with increasing concentrations of the tested antibacterial drug. During subsequent transfers to increasing concentrations, 50 μL was also taken from the last tube, in ascending order, in which visual growth was observed. The selection process was stopped if the MBC had not changed within 6–8 passages.

## 4. Conclusions

Herein, we demonstrated that new pyridinium bis-QACs with aromatic spacers possessed a broader spectrum of antibacterial activity compared to the widely used commercial antiseptics benzalkonium chloride and cetylpyridinium chloride. Hit-QACs showed good bacteriostatic (MICs 0.5 mg/L for Gram-positive *S. aureus*, and 4 to 16 mg/L for Gram-negative bacteria), bactericidal (MBCs 4 mg/L for Gram-positive *S. aureus*, and 8 to 16 mg/L for Gram-negative bacteria), and promising biofilm-eradicating properties (MBECs 8 to 16 mg/L for Gram-positive *S. aureus*, and 16 mg/L for Gram-negative *E. coli*). Moreover, QACs’ biocidal properties against planktonic cells and biofilms greatly improved when used in combination with different alcohols, especially with isopropanol (effective in 78% of the cases for planktonic cells, and in 91% of the cases for biofilms). Bacterial resistance studies showed that the new QACs had a more stable effect against Gram-negative bacteria than the modern sanitizer octenidine dichloride. Thus, **QAC 1** was on the same level of MBC after the month of incubation with *E. coli*. In light of the obtained results, we are encouraged to continue investigations of this biocide class.

## Figures and Tables

**Figure 1 pharmaceuticals-15-00514-f001:**
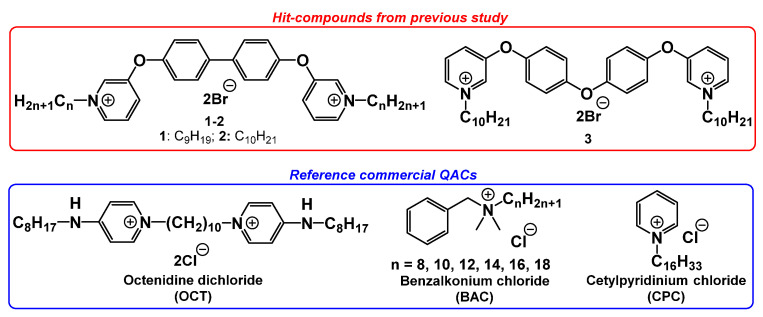
Tested bis-QACs: hit compounds **1**–**2** from ref. [15], and **3** from ref. [16].

**Figure 2 pharmaceuticals-15-00514-f002:**
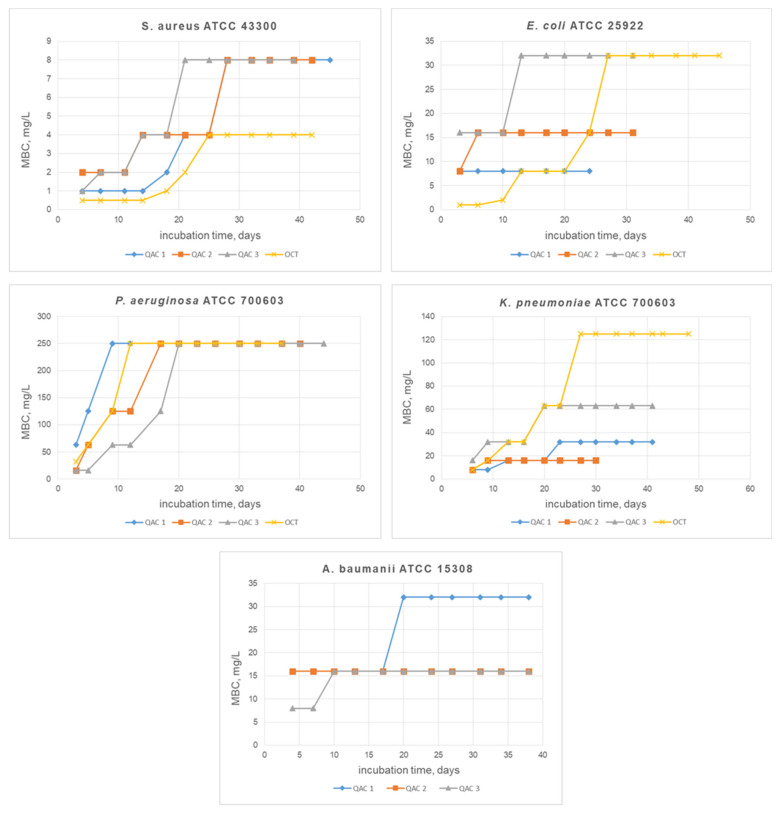
Resistance study. Graphs show the growth of QACs bactericidal concentrations with increasing incubation time on each pathogen.

**Table 1 pharmaceuticals-15-00514-t001:** Intensity of biofilm formation by *reference strains*.

Strains	OD_590_	Intensity
*E. coli* ATCC 25922	0.272 ± 0.011	modest
*K. pneumoniae* ATCC 700603	0.255 ± 0.008	modest
*S. aureus* ATCC 43300	0.301 ± 0.008	modest
*P. aeruginosa* ATCC 27853	0.383 ± 0.013	high
*A. baumannii* ATCC 15308	0.391 ± 0.005	high

**Table 2 pharmaceuticals-15-00514-t002:** Intensity of biofilm formation by *clinical strains*.

Strains	OD_590_	Intensity
*E. coli* B-3421/19	0.287 ± 0.005	modest
*K. pneumoniae* B-2523/18	0.399 ± 0.011	high
*S. aureus* B-8648	0.343 ± 0.022	high
*P. aeruginosa* B-2099/18	0.452 ± 0.018	high
*A. baumannii* B-2996/18	0.415 ± 0.012	high

**Table 3 pharmaceuticals-15-00514-t003:** Antibacterial activity of new bis-QACs against planktonic cells and biofilms.

Compounds	MIC/MBC (mg/L)									
	Reference Strains					Clinical Strains				
	*Sa*	*Ec*	*Kp*	*Ab*	*Pa*	*Sa*	*Ec*	*Kp*	*Ab*	*Pa*
*Planktonic cells*										
**1**: *n* = 9, Br	0.5	4	8	16	16	0.5	4	4	16	16
	4	8	8	16	63	4	16	8	16	16
**2**: *n* = 10, Br	1	8	16	16	16	1	8	16	32	250
	8	8	63	16	16	4	8	63	63	500
**3**: *n* = 10, Br	1	16	16	16	16	1	16	16	16	8
	8	32	16	16	16	4	63	16	16	63
**BAC**	125	4	500	>500	>500	125	500	500	>500	>500
	250	8	500	>500	>500	500	>500	500	>500	>500
**CPC**	4	8	63	32	500	2	8	8	16	32
	16	8	250	125	>500	16	63	16	32	125
**OCT**	0.5	0.5	4	32	8	0.5	2	2	32	32
	2	0.5	8	125	16	2	4	4	125	63
*Biofilms*										
**1**: *n* = 9, Br	8	16	16	32	125	8	16	16	32	125
	16	16	32	125	500	8	32	125	125	500
**2**: *n* = 10, Br	4	16	16	250	125	8	32	16	32	250
	16	16	250	250	>500	16	32	250	500	500
**3**: *n* = 10, Br	8	16	32	63	125	8	16	32	32	250
	16	16	125	500	>500	16	63	250	500	>500
**BAC**	>500	250	>500	>500	>500	>500	>500	>500	>500	>500
	>500	250	>500	>500	>500	>500	>500	>500	>500	>500
**CPC**	16	16	63	250	>500	8	125	63	250	>500
	63	63	250	500	>500	32	250	500	500	>500
**OCT**	4	8	16	250	125	4	16	16	32	250
	8	16	63	250	500	8	125	125	250	250

**Table 4 pharmaceuticals-15-00514-t004:** Antibacterial activity of QACs with different alcohols against planktonic cells.

Compositions	MIC/MBC (mg/L)									
	Reference Strains					Clinical Strains				
	*Sa*	*Ec*	*Kp*	*Ab*	*Pa*	*Sa*	*Ec*	*Kp*	*Ab*	*Pa*
**PhE 1**: n = 9, Br	2	2	4	8	16	2	2	2	16	16
	4	2	8	8	16	4	2	4	16	16
**PhE 2**: n = 10, Br	2	2	2	8	16	2	4	4	16	16
	4	2	4	16	32	4	4	4	16	63
**PhE 3**: n = 10, Br	2	2	4	8	16	2	4	8	16	16
	4	4	4	16	16	4	4	16	16	16
**PhE BAC**	4	16	16	16	32	1	16	32	16	16
	16	16	16	16	63	16	32	32	32	125
**PhE OCT**	1	2	1	2	2	1	2	2	4	2
	8	2	1	2	2	8	4	4	8	8
**PhE/H_2_O** (control)	125	32	63	16	16	250	63	125	16	32
	500	32	125	125	125	500	63	250	125	125
**IPA 1**: n = 9, Br	1	2	2	2	2	1	2	2	2	2
	1	2	2	2	4	1	2	2	2	2
**IPA 2**: n = 10, Br	1	2	4	2	2	1	2	4	2	2
	1	2	4	2	2	2	2	8	2	8
**IPA 3**: n = 10, Br	1	2	2	2	2	1	2	2	2	2
	1	2	4	2	4	2	2	2	2	4
**IPA BAC**	4	8	4	8	32	1	16	32	16	16
	16	8	16	32	63	16	125	32	32	63
**IPA OCT**	1	4	1	2	1	0.5	4	2	2	4
	2	4	1	2	1	2	4	4	4	4
**IPA/H_2_O** (control)	63	32	32	16	16	63	32	32	16	16
	250	32	63	63	63	250	32	125	32	63
**IPAP 1**: *n* = 9, Br	2	2	2	0,5	8	2	2	4	4	8
	4	2	2	1	8	4	2	4	4	8
**IPAP 2**: *n* = 10, Br	2	2	2	2	8	2	4	2	4	8
	4	2	2	2	8	4	4	2	4	8
**IPAP 3**: *n* = 10, Br	2	2	1	2	8	2	2	2	4	8
	2	2	1	8	16	4	2	2	4	8
**IPAP BAC**	4	16	16	16	16	1	16	16	16	16
	16	16	16	16	63	16	16	32	32	63
**IPAP OCT**	1	1	1	2	2	0,5	2	2	2	2
	4	1	1	2	2	4	2	8	4	4
**IPAP/H_2_O** (control)	16	32	63	16	16	63	63	125	16	16
	250	32	125	63	63	250	63	125	63	63

Note: color indicates results that do not differ (or worse) from those of individual substances or control samples, i.e., formulation is not effective.

**Table 5 pharmaceuticals-15-00514-t005:** Antibacterial activity of QACs with different alcohols against biofilms.

Compositions	MIC/MBC (mg/L)									
	Reference Strains					Clinical Strains				
	*Sa*	*Ec*	*Kp*	*Ab*	*Pa*	*Sa*	*Ec*	*Kp*	*Ab*	*Pa*
**PhE 1**: *n* = 9, Br	2	8	16	250	63	4	4	63	125	63
	2	16	32	250	63	4	4	63	125	125
**PhE 2**: *n* = 10, Br	1	4	32	63	32	4	8	63	32	32
	2	16	63	250	63	4	16	63	125	250
**PhE 3**: *n* = 10, Br	2	16	16	125	63	4	1	63	32	32
	2	32	63	500	250	8	16	125	125	125
**PhE BAC**	16	16	32	32	125	8	16	32	32	63
	16	16	32	32	125	32	16	32	32	125
**PhE OCT**	1	4	8	16	16	2	4	16	16	16
	4	8	16	32	63	4	16	32	32	32
**PhE/H_2_O** (control)	32	16	63	63	125	63	32	63	63	32
	63	32	125	250	125	125	63	125	125	125
**IPA 1**: *n* = 9, Br	2	4	8	32	16	2	4	8	32	16
	8	4	8	63	32	8	4	8	63	32
**IPA 2**: *n* = 10, Br	4	4	16	32	16	4	4	16	16	32
	8	4	32	32	32	8	4	63	32	32
**IPA 3**: *n =* 10, Br	2	4	8	32	16	1	4	16	16	16
	4	4	16	32	32	2	4	32	16	63
**IPA BAC**	16	16	32	32	125	16	16	32	32	63
	16	16	32	32	125	32	16	32	32	125
**IPA OCT**	2	4	4	16	8	2	4	16	16	16
	2	8	8	32	32	4	4	32	32	32
**IPA/H_2_O** (control)	32	16	63	63	125	63	32	63	63	32
	63	32	63	250	125	125	63	125	250	125
**IPAP 1**: *n* = 9, Br	1	4	8	32	16	2	1	16	32	16
	2	4	16	63	63	4	4	32	32	63
**IPAP 2**: *n* = 10, Br	1	8	16	32	32	2	8	32	32	32
	2	32	63	125	63	4	8	63	63	125
**IPAP 3**: *n* = 10, Br	4	16	16	32	16	2	2	32	32	16
	4	16	16	63	63	4	4	32	63	63
**IPAP BAC**	16	1	32	32	125	16	1	32	32	63
	16	2	32	63	250	32	8	32	32	500
**IPAP OCT**	1	2	8	8	16	2	2	16	16	16
	4	4	8	16	63	4	8	16	32	125
**IPAP/H_2_O** (control)	32	16	63	63	125	32	32	63	63	32
	63	32	63	125	125	125	63	125	125	125

Note: color indicates results that do not differ (or worse) from those of individual substances or control samples, i.e., formulation is not effective.

**Table 6 pharmaceuticals-15-00514-t006:** Effectivity of antiseptic compositions.

Compositions	Effectivity										
	Reference Strains					Clinical Strains					Σ
	*Sa*	*Ec*	*Kp*	*Ab*	*Pa*	*Sa*	*Ec*	*Kp*	*Ab*	*Pa*	
*Planktonic cells*											
**PhE**	40%	80%	90%	70%	40%	20%	80%	70%	40%	70%	60%
**IPA**	50%	80%	100%	100%	80%	60%	80%	70%	80%	80%	**78%**
**IPAP**	50%	80%	100%	90%	80%	30%	90%	70%	90%	80%	**76%**
*Biofilms*											
**PhE**	100%	50%	70%	40%	90%	100%	100%	60%	60%	20%	69%
**IPA**	90%	90%	90%	90%	80%	100%	100%	90%	100%	80%	**91%**
**IPAP**	100%	70%	90%	80%	80%	100%	100%	60%	70%	50%	80%

## Data Availability

Data is contained within the article.

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
