# Peer review of "Microbiological Evaluation of Novel Bis-Quaternary Ammonium Compounds: Clinical Strains, Biofilms, and Resistance Study"

_pharmaceuticals, 2022, doi:10.3390/ph15050514_

Round 1

Reviewer 1 Report

This short and basic report aims at determining the antibacterial activities of a number of novel quaternary ammonium compounds (QACs) against clinically relevant Gram-negative and -positive bacteria. It also attempts to assess the antiseptic and anti-biofilm properties of the same compounds alone or in combination with various alcohols. My comments are as follows:

Major comments

* Since potent pyridinium bis-QACs with aromatic spacers were designed in a previous study by the authors, the present study is a quick and simple in vitro evaluation of these agents, much like an MSc study. Therefore, the amount of data required for an "Article" is not achieved. This manuscript can only be published as in the form of a "Comment" in Pharmaceuticals. Please reorganize your manuscript accordingly.

* In Introduction, the first paragraph necessitates proper citations. For example, "Bacteria in biofilms are known to have increased survival rate in presence of aggressive substances, immune defense factors, and antibacterial drugs." is not a general-knowledge-based sentence.

* Please rephrase the sentence "Herein, we demonstrated that new pyridinium bis-QACs with aromatic spacer possessed a broad spectrum of high antibacterial activity compared to the widely known commercial antiseptics benzalkonium chloride and cetylpyridinium chloride." in Conclusion. When comparing two things, try and use comparative adjectives such as "broader" and "higher".

* Subsection 3.2 in Materials and Methods (i.e. Identification of microorganisms) should be detailed.

* The manuscript was written with poor English (both grammar and punctuation).

Minor comments

* gram-negative/positive -> Gram-negative/positive

* be consistent with "litre" - mg/l or mg/L?

* genus and species names must be written in italics

* separate "degree Celcius" from the associated numeric value

* use a figure dash (not a dash) when expressing a numeric range

* percent concentrations should be expressed v/v or w/v

Author Response

* Since potent pyridinium bis-QACs with aromatic spacers were designed in a previous study by the authors, the present study is a quick and simple in vitro evaluation of these agents, much like an MSc study. Therefore, the amount of data required for an "Article" is not achieved. This manuscript can only be published as in the form of a "Comment" in Pharmaceuticals. Please reorganize your manuscript accordingly.

Answer:

Manuscript was reorganized in “Communication” format according with comment.

* In Introduction, the first paragraph necessitates proper citations. For example, "Bacteria in biofilms are known to have increased survival rate in presence of aggressive substances, immune defense factors, and antibacterial drugs." is not a general-knowledge-based sentence.

Answer:

The introduction was supplemented with following citations according with comment: 1.           O’NEILL, J. Tackling Drug-Resistant Infections Globally: final report and recommendations; London, United Kingdom, 2016; p 84. 2.            Flemming, H.-C.; Wingender, J.; Szewzyk, U.; Steinberg, P.; Rice, S.A.; Kjelleberg, S. Biofilms: an emergent form of bacterial life. Nature Reviews Microbiology 2016, 14, 563-575, doi:10.1038/nrmicro.2016.94. 3. Harriott, M.M. Biofilms and Antibiotics. In Reference Module in Biomedical Sciences, Elsevier: 2019; https://doi.org/10.1016/B978-0-12-801238-3.62124-4. 4. Hall, C.W.; Mah, T.-F. Molecular mechanisms of biofilm-based antibiotic resistance and tolerance in pathogenic bacteria. FEMS Microbiology Reviews 2017, 41, 276-301, doi:10.1093/femsre/fux010. 7. Gilbert, P.; Moore, L.E. Cationic antiseptics: diversity of action under a common epithet. Journal of Applied Microbiology 2005, 99, 703-715, doi:https://doi.org/10.1111/j.1365-2672.2005.02664.x. 8.Tischer, M.; Pradel, G.; Ohlsen, K.; Holzgrabe, U. Quaternary Ammonium Salts and Their Antimicrobial Potential: Targets or Nonspecific Interactions? ChemMedChem 2012, 7, 22-31, doi:https://doi.org/10.1002/cmdc.201100404.

* Please rephrase the sentence "Herein, we demonstrated that new pyridinium bis-QACs with aromatic spacer possessed a broad spectrum of high antibacterial activity compared to the widely known commercial antiseptics benzalkonium chloride and cetylpyridinium chloride." in Conclusion. When comparing two things, try and use comparative adjectives such as "broader" and "higher".

Answer:

Sentence was rephrased according with comment. New variant: “Herein, we demonstrated that new pyridinium bis-QACs with aromatic spacer possessed a broader spectrum of antibacterial activity compared to the widely known commercial antiseptics benzalkonium chloride and cetylpyridinium chloride”

* Subsection 3.2 in Materials and Methods (i.e. Identification of microorganisms) should be detailed.

Answer:

Subsection 3.2 was supplemented with a more detailed methodology for the identification of microorganisms (see page 9)

Minor comments

* gram-negative/positive -> Gram-negative/positive

* be consistent with "litre" - mg/l or mg/L?

* genus and species names must be written in italics

* separate "degree Celcius" from the associated numeric value

* use a figure dash (not a dash) when expressing a numeric range

* percent concentrations should be expressed v/v or w/v

Answer:

All highlighted errors were corrected.

Reviewer 2 Report

In the present work Nikita et al had evaluated the antimicrobial properties of previously identified hit Bis-Quaternary Ammonium Compounds. The work is performed using sound protocols and is suitable for publication in pharmaceuticals journal after addressing the following.

  1. Abstract and conclusion need improvement. Specifically add the Minimal inhibitory concentrations (MICs) and minimal bactericidal concentrations (MBCs) as well as minimum biofilm inhibitory concentrations (MBICs) and minimum biofilm eradication concentrations (MBECs) values of the most potent compounds.
  2. The names of the microbial species should be in italics throughout the manuscript.
  3. Minor English language correction is required.

Author Response

1. Abstract and conclusion need improvement. Specifically add the Minimal inhibitory concentrations (MICs) and minimal bactericidal concentrations (MBCs) as well as minimum biofilm inhibitory concentrations (MBICs) and minimum biofilm eradication concentrations (MBECs) values of the most potent compounds.

Answer:

Abstract (see page 1) and conclusion (see page 11) were improved according to the comment.

2. The names of the microbial species should be in italics throughout the manuscript.

Answer:

Corrected accordingly.

3. Minor English language correction is required.

Answer:

All highlighted errors were corrected. We thank the reviewer for the comments!

Reviewer 3 Report

Frolov et al, submitted title, "Microbiological Evaluation of Novel Bis-Quaternary Ammonium Compounds: Clinical Strains, Biofilms, and Resistance Study" where various antimicrobial assays were performed.

However, there are some points which needs to be addressed.

  1. There is a vast literature on antibacterial/ antimicrobial activity of quaternary ammonium compounds, how the current study is different, please elaborate.
  2. There must be some explanation why current compounds have superiority or improved activity than previous reported potent ammonium quaternary compounds. For this, authors needs to compare their compounds/one of most potent-one with at-least one previous reported ammonium quaternary compound.
  3. There is little-to-none explanation provided by the authors about the mechanism of these compounds. Please provide sufficient information.
  4.  In literature, it is well known that these kind of quaternary compounds show poor pharmacokinetics, therefore such molecules lack cell selectivity and commonly fail in clinical studies. Authors needs to highlight these issues either by performing additional experiments or at-least highlights the points based on current results.

As this manuscript is under the section of "Medicinal chemistry" in a special issue " Drug candidates for treatment of infectious diseases" , therefore authors needs to incorporate either the structure activity relationship or other medicinal chemistry tools. 

Although, manuscript certainly have some merit, but doesn't provide sufficient justification about these compounds medicinal chemistry, which needs to be addressed along with other points raised in above comments before considering in the current journal.

Author Response

1. There is a vast literature on antibacterial/ antimicrobial activity of quaternary ammonium compounds, how the current study is different, please elaborate.

Answer:

This study was carried out on new QAСs structures obtained in our laboratory. The work allows to evaluate the effect of new aromatic spacers (biphenyl and diphenyl ether) on activity against clinical strains, biofilms and the formation of bacterial resistance. For example, some new insights into bacterial resistance were discovered, that aromatic moieties could reduce recognition by resistance system in Gram-negative bacteria. Moreover, this study is not only investigates activity against planktonic bacterial cells (as usual work), but also against biofilms and clinically isolate pathogens, and addresses combine action of QACs with different alcohols.

2. There must be some explanation why current compounds have superiority or improved activity than previous reported potent ammonium quaternary compounds. For this, authors needs to compare their compounds/one of most potent-one with at-least one previous reported ammonium quaternary compound.

Answer:

Presented work is a continuation of these two studies (Vereshchagin, A.N.; Gordeeva, A.M.; Frolov, N.A.; Proshin, P.I.; Hansford, K.A.; Egorov, M.P. Synthesis and Microbiological Properties of Novel Bis-Quaternary Ammonium Compounds Based on Biphenyl Spacer. European Journal of Organic Chemistry 2019, 2019, 4123-4127, doi:https://doi.org/10.1002/ejoc.201900319; Vereshchagin, A.N.; Frolov, N.A.; Konyuhova, V.Y.; Hansford, K.A.; Egorov, M.P. Synthesis and microbiological properties of novel bis-quaternary ammonium compounds based on 4,4′-oxydiphenol spacer. Mendeleev Communications 2019, 29, 523-525, doi:https://doi.org/10.1016/j.mencom.2019.09.015) of our laboratory. They provide a detailed comparison (as well as stricture/activity relationship) of the selected hit-compounds (QAC 1-3) with previously obtained QACs.

3. There is little-to-none explanation provided by the authors about the mechanism of these compounds. Please provide sufficient information.

Answer:

Information about mechanism of QACs was added in introduction part with correlated citations (see page 2). 

4.  In literature, it is well known that these kind of quaternary compounds show poor pharmacokinetics, therefore such molecules lack cell selectivity and commonly fail in clinical studies. Authors needs to highlight these issues either by performing additional experiments or at-least highlights the points based on current results.

Answer:

Foremost, QACs is commonly used as disinfectants or antiseptics, i.e. not for internal use. Therefore, pharmacokinetics is not the main characteristic in our case. However, we have planned preclinical studies of such QACs. Concerning cell selectivity, in abovementioned work (see comment 2) cytotoxicity study showed, that hit-QACs mostly active in concentrations below toxic values for human embryonic kidney cells and human red blood cells.

As this manuscript is under the section of "Medicinal chemistry" in a special issue " Drug candidates for treatment of infectious diseases" , therefore authors needs to incorporate either the structure activity relationship or other medicinal chemistry tools. 

Answer:

SAR study is represented in abovementioned work (see comment 2).

Although, manuscript certainly have some merit, but doesn't provide sufficient justification about these compounds medicinal chemistry, which needs to be addressed along with other points raised in above comments before considering in the current journal.

Answer:

Synthesis of QAC 1-3 is represented in abovementioned work (see comment 2). Current study is more focused on biological research of these compounds. However, we think, that our investigation is fulfils the requirements of special issue "Drug candidates for treatment of infectious diseases". The whole purpose of QACs is combating bacterial pathogens, and our compounds showed better biological properties than commercial ones. Moreover, new bis-QACs is much easier to acquire with two-step synthetic route.

Round 2

Reviewer 3 Report

The authors made several changes and addressed various points raised in the previous evaluation. As a result, the publication has enough elements to fulfill the criteria to consider in the special issue, "Drug Candidates for the Treatment of Infectious Diseases."